# SiO_2_ Fibers of Two Lengths and Their Effect on Cellular Responses of Macrophage-like Cells

**DOI:** 10.3390/molecules27144456

**Published:** 2022-07-12

**Authors:** Denisa Smela, Chia-Jung Chang, Ludek Hromadko, Jan Macak, Zuzana Bilkova, Akiyoshi Taniguchi

**Affiliations:** 1Department of Biological and Biochemical Sciences, Faculty of Chemical Technology, University of Pardubice, Studentska 573, 532 10 Pardubice, Czech Republic; zuzana.bilkova@upce.cz; 2Research Center for Functional Materials, National Institute for Materials Science, 1-1 Namiki, Tsukuba 305-0044, Japan; annachang0128@gmail.com (C.-J.C.); taniguchi.akiyoshi@nims.go.jp (A.T.); 3Center of Materials and Nanotechnologies, Faculty of Chemical Technology, University of Pardubice, Nam. Cs. Legii 565, 530 02 Pardubice, Czech Republic; ludek.hromadko@upce.cz (L.H.); jan.macak@upce.cz (J.M.); 4Central European Institute of Technology, Brno University of Technology, Zerotinovo nam. 617/9, 601 77 Brno, Czech Republic

**Keywords:** SiO_2_ nanofibers, cytotoxicity, immunoreactivity, THP-1-derived macrophage-like cells

## Abstract

The immunoreactivity or/and stress response can be induced by nanomaterials’ different properties, such as size, shape, etc. These effects are, however, not yet fully understood. This study aimed to clarify the effects of SiO_2_ nanofibers (SiO_2_NFs) on the cellular responses of THP-1-derived macrophage-like cells. The effects of SiO_2_NFs with different lengths on reactive oxygen species (ROS) and pro-inflammatory cytokines TNF-α and IL-1β in THP-1 cells were evaluated. From the two tested lengths, it was only the L-SiO_2_NFs with a length ≈ 44 ± 22 µm that could induce ROS. Compared to this, only S-SiO_2_NFs with a length ≈ 14 ± 17 µm could enhance TNF-α and IL-1β expression. Our results suggested that L-SiO_2_NFs disassembled by THP-1 cells produced ROS and that the inflammatory reaction was induced by the uptake of S-SiO_2_NFs by THP-1 cells. The F-actin staining results indicated that SiO_2_NFs induced cell motility and phagocytosis. There was no difference in cytotoxicity between L- and S-SiO_2_NFs. However, our results suggested that the lengths of SiO_2_NFs induced different cellular responses.

## 1. Introduction

Nanomaterials’ (NMs) interactions with cells depend on the physicochemical properties of NMs, such as chemical composition, size, shape, and surface area or concentration [1,2]. Moreover, the surface characteristics, such as charge [3], hydrophobicity [4], protein corona [5] and related gravy index score [6,7], type of coatings [8,9], ligands [10], etc., play a significant role in various biological processes, such as protein adsorption, interaction with biological membranes, cellular uptake, and immune responses.

Regarding the size of NMs, it is predictable that various human cell types tend to internalize smaller sizes more easily than the larger ones [11,12] and that the different sizes can induce different cellular responses. For example, with gold nanoparticles, the sizes <1.4 nm that are internalized by the diffusion process are highly cytotoxic, while the larger nanoparticles around 15 nm are completely nontoxic [13]. In a study focused on the uptake and interaction between cells and NMs (sizes 0.5–5 µm), it was also proposed that a general interaction can be weaker, with smaller NMs compared to the bigger NMs [14].

Based on the morphological similarity of nanofibers’ shapes to pathogenic fibers such as asbestos, there is concern about safety and health implications after exposure [15]. Many studies testing new NMs have confirmed this. Hamilton et al. (2009) compared anatase TiO_2_ in the form of nanospheres and two different lengths of nanobelts. They described that the cytotoxicity of nanospheres and short nanobelts was insignificant, while the nanobelts with lengths >15 µm showed a significant increase in toxicity [16]. Schinwald et al. (2012) suggested that, for silver nanowires, the threshold length for the pathogenicity of the material in mice is >5 µm [17]. On the other hand, Cacchioli et al. (2014) described the cytocompatibility of SiC/SiO_2_ core–shell nanowires with different cell types up to a concentration of 100 µg/mL [18].

Understanding the complexity of the effect that NMs have on cells (in vitro and in vivo) is crucial for the material to be safely and effectively used in various applications. Therefore, it is necessary to pay attention to the verification of declared properties, which is the only way to validate their overall biological impact.

Great potential is attributed to biocompatible materials such as SiO_2_ [19]. This material has been studied for several medical applications, such as, e.g., wound dressing, where SiO_2_ nanofibers with immobilized tetracycline were used for their antibacterial activity [20]. Wang et al. (2021) used SiO_2_ nanofibers to mimic the colorectal microenvironment to improve the current in vitro assessment of the therapy of colorectal cancer [21]. SiO_2_ nanofibers [22], nanoparticles [23], and nanotubes [24] were tested as carriers for drug delivery systems. The characteristics of SiO_2_ fibers, such as high mechanical strength, low cytotoxicity, and biodegradability, also seem to be very promising for bone tissue engineering and designing new scaffolds [25]. To assess the toxicological hazard of nanofibers for human cells in their complexity, we evaluated the effect of SiO_2_ nanofibers (SiO_2_NFs) differing in lengths on THP-1-derived macrophage-like cells as a representative of immunocompetent cells. In order to determine the effects of SiO_2_NF lengths on cellular responses, we evaluated the cell viability, as well as ROS and cytokine production. In addition, we studied the differences in the expression of F-actin. The results of this study can help us gain insight into the cytotoxicity and the immunoreactivity of this fibrous material.

## 2. Results

This work aimed to evaluate the biocompatibility/toxicity of SiO_2_NFs and to determine in what manner they interact with immunocompetent cells. Alongside the viability tests, we focused on the immunoreactivity of the material, whether the shape and length can influence the activity of THP-1-derived macrophage-like cells. Monocytes and macrophages are a part of the innate immune system, with a major role in the recognition of foreign pathogens (e.g., nanomaterials), elimination of those pathogens, and also the production of proinflammatory cytokines. THP-1 cells are a well-established and suitable in vitro model system for studying the modulation of monocyte and macrophage functions. Therefore, the results obtained from tests with THP-1 cells can hint at potential immune responses in vivo [26]. The tested nanofibers showed in Figure 1 were SiO_2_-based, differing in lengths (longer L-SiO_2_NFs of ≈44 µm ± 22 µm and shorter S-SiO_2_NFs of ≈14 µm ± 17 µm).

The cell viability of THP-1-derived macrophage-like cells was tested with the WST-1 assay after 24 and 48 h of exposure to both lengths of SiO_2_NFs (Figure 2). After 24 h of treatment, there was no significant change in the cell viability (Figure 2A). Compared to the nontreated control cells, the cell viability after 48 h of treatment increased (20–30%) for the cells treated with lower doses of SiO_2_NFs (up to 100 µg/mL). At the highest concentration of SiO_2_NFs (200 µg/mL), the cell viability was significantly lowered (25%, Figure 2B). There was no difference in cytotoxicity between L- and S-SiO_2_NFs. The cell viability data suggested that the SiO_2_NF lengths would not have an important effect on the induced cytotoxic response.

However, these first results led us to an additional detailed study where we tested the cellular responses to different lengths of SiO_2_NFs. The results monitoring reactive oxygen species (ROS) production induced by SiO_2_NFs are summarized in Figure 3. ROS are considered to be an important mediator in proinflammatory signaling pathways activation, and the increase in their production is connected to phagocytosis or cell stimulation with various agents [27]. Figure 3 shows the experimental data obtained by the H_2_DCFDA assay after 6 and 24 h of treatment with different concentrations of L- and S-SiO_2_NFs (10, 100, and 200 µg/mL), as well as with lipopolysaccharide (LPS) (1 µg/mL) as a positive control. LPS is a component of Gram-negative bacteria membranes that stimulates macrophages, resulting in an inflammatory response, including ROS production. Within 6 h of treatment, the cells treated with both SiO_2_NFs showed no significant difference in ROS production compared to the nontreated control cells (Figure 3A). The opposite trend was observed in cells after 24 h of treatment (see Figure 3B). The ROS production after 24 h of treatment with L-SiO_2_NFs (10 µg/mL) was significantly increased. After 24 h of exposure to 100 and 200 µg/mL of L-SiO_2_NFs, the ROS production was at the same level as the nontreated control cells, which might be due to the cell damage caused by high L-SiO_2_NF concentrations. The ROS production levels after exposure to all concentrations of S-SiO_2_NFs were comparable to the nontreated control cells’ level. Notable results were obtained for the LPS treatment (positive control) after 24 h, where the ROS level decreased to the nontreated control cells’ level. This result was in agreement with the data presented by Widdrington et al. (2018), who measured the LPS-induced ROS production in THP-1 cells for up to 72 h and observed a peak in ROS production at the 6-h mark and then a gradual decrease afterwards. They suggested it was the result of compensatory cellular mechanisms (including antioxidant defenses, mitochondrial biogenesis, and mitophagy) triggered after the LPS treatment [28]. The results for the materials tested in this study indicated that ROS production could be induced only by L-SiO_2_NFs after 24 h of incubation.

The proinflammatory cytokines such as TNF-α and IL-1β are required for activation of the innate immune response and mediation of the recruitment, activation, and adherence of circulating phagocytic cells (macrophages and neutrophils) [29]. In the following experiments, we studied the ability of L- and S-SiO_2_NFs to increase TNF-α and IL-1β production. The levels of TNF-α and IL-1β produced by treated THP-1-derived macrophage-like cells are graphically expressed in Figure 4A and Figure 4B, respectively. After the treatment with L-SiO_2_NFs, there was only a small increase in cytokine production for the highest tested concentration of L-SiO_2_NF. On the other hand, the cellular response to S-SiO_2_NF was much more prominent. The highest concentration of S-SiO_2_NF (200 µg/mL) caused an increase in both measured cytokines, especially in IL-1β production. The results indicated that TNF-α and IL-1β expression and, thus, inflammation is concentration-dependent and can be induced by S-SiO_2_NFs but not by L-SiO_2_NFs.

Following this, the amount and localization of F-actin were studied to determine the effect of SiO_2_NFs on phagocytic activity and cell motility. The image analysis of the F-actin signal after staining (Figure 5) revealed that the measured signal per cell was similar for the nontreated control cells and LPS-treated cells, while the cells treated with SiO_2_NFs showed a lower signal per cell. The most prominent difference was in cells treated with L-SiO_2_NFs (80% compared to the nontreated control cells), while the signal per cell in cells treated with S-SiO_2_NFs was at a similar level to the nontreated control cells (95%). However, there was a clear difference in distribution of the stained F-actin within the cells treated with SiO_2_NFs. In the nontreated control cells, F-actin was mostly homogeneously distributed, with only a few F-actin bundles—podosomes. Those structures are typically formed in monocyte-derived macrophages [30]. We saw F-actin presented mostly in the form of podosomes, especially within the cells treated with L-SiO_2_NFs (which also had the lowest measured signal per cell). This suggested that SiO_2_NFs, especially L-SiO_2_NFs, caused an increase in cell motility and phagocytosis.

## 3. Discussion

The impulse for this study was a new type of nanofiber based on SiO_2_, prepared by centrifugal spinning, an alternative technique to traditional electrospinning [31]. Nanofibers prepared by either centrifugal spinning [32] or electrospinning [33] have recently attracted significant attention due to their unique properties and morphologies that can be utilized in biomedicine, filtration, catalysis, and many other fields. In medicine, nanofibers are suitable for bone tissue engineering [34], 3D cell growth [35] drug delivery [36], and bioseparation [37] but also as templates for the deposition of other materials [38,39]. Centrifugal spinning offers a number of advantages compared to electrospinning, where the most interesting one for clinical applications is the use of spun solutions without any toxic elements in their preparation [31].

These fibers have great potential in biomedicine, but in order to be successful, it is important to evaluate their bioreactivity more thoroughly than only with the commonly used panel of methods for cytotoxicity testing. It was proven several times in the past that even biocompatible materials can cause adverse effects such as immunotoxicity, genotoxicity, hepatotoxicity, etc. Therefore, we should also attempt to evaluate the influence of newly developed materials on specialized cells such as immune system cells. Our aim was to understand how the immune system, especially THP-1-derived macrophage-like cells, respond to treatment with these fibers.

The THP-1 cell line is widely used as a model to study the immune function/response [26]. THP-1 cells acquire a macrophage-like phenotype and functional characteristics by PMA treatment [40]. Cell adhesion, spreading, and an increased cytoplasmatic volume are accepted hallmarks of differentiation into macrophages. According to our optimized protocol, THP-1 cells were well-differentiated after 48 h of incubation with 50 ng/mL PMA, followed by a 24-h rest period. After differentiation, the THP-1 cells became adherent with a lower proliferation rate. In agreement with the literature [41,42], the increase in the CD14 marker on the cell surface was another sign of completed differentiation (Appendix A in the Appendix A).

As was confirmed by our results, the length of the fiber had a significant effect on the cellular response. Ideally, the fiber preparation technique should allow to repeatedly prepare fibers with a defined length for them to be used for a specific application. Next, the post-synthetic modifications should also be taken into account and tested to determine whether they change the final cellular response to the material. We performed our experiments with the SiO_2_ nanofibers of different lengths after milling. The shortening of the fibers was observed after as little as 5 s of milling (see the difference between Figure 1A,B) when the average fiber length was ≈ 44 ± 22 µm. After milling for 60 s (Figure 1C), we obtained a batch with the majority of fibers with a length of ≈ 14 ± 17 µm. Even though the NF lengths were not completely uniform, we were still able to prepare aliquots with defined average lengths with different milling times. SiO_2_NFs prepared this way were easy to work with in a dry form, as well as in aqueous solutions without any spontaneous aggregation. After dispersing, the nanofibers settled gradually but homogenous suspension was easy to achieve again with only a few seconds of vortexing or thorough mixing.

As we presented in this work, the lengths of the tested SiO_2_NFs did not have any effect on cell viability, and they proved to be nontoxic up to the concentration 100 µg/mL even after 48 h of exposure (Figure 2). Moreover, after 48 h, there was a significant increase in cell viability compared to the nontreated control cells, with the exception of the highest tested concentration (200 µg/mL). These findings suggest a higher biocompatibility of SiO_2_, which contrasts with toxic TiO_2_ nanobelts, lengths >15 µm [16], or silver nanowires, lengths > 5 µm [17], mentioned before.

However, we observed the apparent differences in the cellular response based on the lengths of SiO_2_NFs in other performed assays. S-SiO_2_NFs did not cause any significant increase in ROS production, but there was an increase in ROS production caused by the lowest concentration of L-SiO_2_NFs (Figure 3). There was, however, a significant proinflammatory response to S-SiO_2_NFs at a concentration 200 µg/mL in the form of increased TNF-α and IL-1β secretion after 24 h of treatment (Figure 4). We also measured the levels of IL-6, but no response was detected (data not shown). It was apparent that, while SiO_2_NFs (especially S-SiO_2_NFs) induced the production of TNF-α and IL-1β, they do not have any effect on IL-6 expression (data not shown).

Lopes et al. (2017) showed similar data for nanofibrillated cellulose [43]. They found no significant cytotoxicity or increase in ROS production but recorded significantly increased TNF-α and IL-1β production after treatment with nanofibrillated cellulose at concentrations > 250 µg/mL. They suggested that this proinflammatory response is driven by the surface chemistry between the cell and nanofibers that are not internalized. Boonrungsiman et al. (2017) also observed the absence of ROS production and differences in cytokine (IL-8 and TNF- α) production in THP-1 cells treated by TiO_2_ NFs (25 µg/mL, lengths 54 ± 32 nm and 73 ± 21 nm) [44]. It was also suggested that the size of asbestos fibers necessary to induce a significant macrophage response is 4 µm, while, for polyester fibers, sizes less than 70 µm induce a significant response [45]. Ye et al. (1999) demonstrated in their study on glass fibers that fibers 17 µm in length induced a more potent response in TNF-α production than their shorter fibers with 7 µm [46]. This is similar to our results obtained after the treatment with S-SiO_2_NFs (14 µm). Consistent with our data was also Padmore et al. (2017), whose results showed that their longer fibers (size 39.3 µm) were able to induce an inflammatory response, including ROS production, unlike their shorter fibers (size 7 µm) [47].

F-actin is, in general, connected to the activation of phagocytosis and the mobility of THP-1-derived macrophage-like cells [48]. Podosomes play a role in an innate immunity, and their functions within macrophages are adhesion and mobility [30]. The difference in signals provided by F-actin staining (Figure 5) confirmed our expectations that the macrophages internalize S-SiO_2_NFs more easily than L-SiO_2_NFs. The character of the interaction between the cells and fibers is determined by their length, and the inability of cells to internalize L-SiO_2_NFs due to its size directly influences the final cellular response.

Our results showed that both lengths of SiO_2_NFs induced an inflammatory response, although they might differ in mechanism. Even though we confirmed our expectations of the minimal cytotoxicity of SiO_2_NFs, it was apparent that the length and concentration of the tested NFs were crucial factors to consider in the evaluating process of the immunocompetent phagocytic cells’ reactivity.

## 4. Materials and Methods

### 4.1. Cell Culture and Monocyte Differentiation

The human myelogenous leukemia cell line THP-1 was obtained from the American Type Culture Collection (Manassas, VA, USA). The cells were stored in RPMI-1640 medium (Nacalai Tesque, Kyoto, Japan) with supplemented 10% heat-inactivated fetal bovine serum (FBS, Corning Life Sciences, Corning, NY, USA), and 100 U/mL penicillin/streptomycin (Nacalai Tesque, Kyoto, Japan) and incubated in a humidified incubator at 37 °C, 5% CO_2_.

THP-1 monocytes at concentration 3 × 10^5^ cells/mL were differentiated into macrophage-like cells by treatment with 50 ng/mL phorbol-12-myristate-13-acetate (PMA, Sigma Aldrich, St. Louis, MO, USA) for 48 h. After differentiation, the cells were washed with a phosphate saline buffer (PBS, pH 7.4, Sigma-Aldrich, St. Louis, MO, USA), refed with fresh RPMI-1640 medium without PMA and left for 24 h, allowing them to recover. Differentiation was verified by evaluating the cell adhesion and morphology under an optical microscope, as well as with CD14 immunostaining.

### 4.2. Nanomaterials

SiO_2_ nanofibers were prepared by centrifugal spinning, according to a previously published recipe [31]. In brief, a solution containing polyvinylpyrrolidone and tetraethyl orthosilicate was used to prepare precursor fibers, which, after spinning, were annealed using an optimized profile for pure SiO_2_. However, the final product had a 3D structure (similar to cotton), which is not suitable for cell tests. Thus, the fibers with a diameter of ≈300 nm were shortened by ball milling using a Fritsch Spartan ball miller. The morphological characterization of the original, as well as the ball-milled fibers, was carried out using a field emission scanning electron microscope (FE-SEM, JSM7500F, JEOL). The original, as well as resulting fibers, are shown in Figure 1. Two fiber lengths were prepared by different durations of milling: the longer nanofibers were obtained by milling for 5 s (L-SiO_2_NFs), while the shorter nanofibers were obtained by milling for 60 s (S-SiO_2_NFs). The average fiber length was ≈44 µm (SD ≈ 22 µm) for L-SiO_2_NFs and ≈ 14 µm (SD ≈ 17 µm) for S-SiO_2_NFs, respectively.

### 4.3. Cytotoxicity Assays

The cytotoxicity of SiO_2_NFs was tested with the WST-1 assay (Sigma Aldrich, St. Louis, MO, USA), which is based on the bioreduction of the tetrazolium salt WST-1 to formazan, where the final absorbance correlates with the number of viable cells. THP-1 cells were seeded in a clear 96-well plate with a flat bottom, 3 × 10^4^ cells per well, in a medium containing PMA. The THP-1 monocytes were differentiated, as described above. After the recovery period, cells were washed and incubated with SiO_2_NFs in RPMI-1640 medium at concentrations 0, 10, 40, 100, and 200 µg/mL. After 24 or 48 h and incubation at 37 °C, 5% CO_2_, the WST-1 reagent was added according to the manufacturer’s instructions, and the well plate was incubated in 37 °C, 5% CO_2_ for 90 min. The absorbance was measured at 450 nm with a Spark™ 10 M multimode microplate reader (Tecan, Männedorf, Switzerland).

### 4.4. ROS Production Assay

For ROS monitoring, the H_2_DCFDA assay (ThermoFisher Scientific, Waltham, MA, USA) was used. Nonfluorescent H_2_DCFDA was converted by oxidation into fluorescent 2′,7′-dichlorofluorescein (DCF), which is used as an indicator of ROS production. THP-1 cells were seeded in a black 96-well plate with a flat bottom, 3 × 10^4^ cells per well, in a medium containing PMA and differentiated as described above. After the recovery period, the cells were washed and incubated with LPS, and resuspended SiO_2_NFs in the medium at concentrations of 0, 10, 100, and 200 µg/mL for 6 or 24 h at 37 °C, 5% CO_2_. The supernatant was then discarded, and the cells were washed. The reagent was diluted in PBS to a final concentration of 10 µM and added to each well. After 15 min of incubation, the fluorescence intensity was measured with a Spark™ 10 M multimode microplate reader (Tecan, Männedorf, Switzerland).

### 4.5. Cytokine Production Assay

For the evaluation of cytokine production (IL-1β and TNF-α), ELISA kits by R&D Systems (Minneapolis, MN, USA) were used. Cells were seeded in a 96-well plate and differentiated with the same protocol as described above. After 24 h of incubation with SiO_2_NFs, the supernatant samples were transferred into the wells of the ELISA kit with the appropriate antibodies (anti-IL-1β and anti-TNF-α). The assay was performed according to the manufacturer’s instructions. The final absorbance was measured at 450 nm with a Spark™ 10 M multimode microplate reader (Tecan, Männedorf, Switzerland).

### 4.6. F-Actin Staining and Confocal Microscopy

Cells were seeded at 4 × 10^4^ per compartment in a CELLview^TM^ dish with a glass bottom in a medium containing PMA. THP-1 monocytes were differentiated as described above. Afterwards, cells were washed with PBS and incubated with SiO_2_NFs. Following the treatment, the medium was removed, and the cells were fixed for 20 min and permeabilized with 0.05% saponin for 10 min. Cells were washed, and 3% bovine serum albumin (BSA) was added for blocking (1 h). The cells were then washed and stained with 1:500 rhodamine-conjugated phalloidin (Life Technologies, Gaithersburg, MD, USA). After 1 h of incubation and 2× washing, the nucleus was stained with 1:300 2-[4-(aminoiminomethyl)phenyl]-1H-Indole-6-carboximidamide hydrochloride (DAPI, Abcam, Cambridge, UK) for 15 min. Cells were incubated at room temperature and then washed again. The images were acquired with a Zeiss LSM 510 META confocal microscope system (Carl Zeiss, Jena, Germany). The image analysis to measure the fluorescence intensity was performed by ImageJ software.

## 5. Conclusions

We showed that the length of centrifugally spun and artificially shortened SiO_2_NFs did not significantly affect the cell viability of THP-1-derived macrophage-like cells. SiO_2_NFs seem to be biocompatible up to the concentration of 100 µg/mL, which is high enough to allow for practical clinical applications that commonly use lower doses. Our results showed that the SiO_2_NF different lengths lead to a different mechanism of macrophage activation. The data suggested that the L-SiO_2_NFs are able to induce a more prominent stress response (ROS production and F-actin localization into podosomes) in THP-1-derived macrophage-like cells. The data obtained for the S-SiO_2_NFs (increased TNF-α and IL-1β production) indicated a proinflammatory effect. Our results suggested that L-SiO_2_NFs disassembled by THP-1 cells produced ROS and that the inflammatory reaction was induced by the uptake of S-SiO_2_NFs by THP-1 cells. Our findings suggest that there is a great potential in SiO_2_NFs as a tool in clinical applications; however, because of the increasing abundance of nanofibers in everyday life and the great diversity between them, we have to adjust the panel of necessary tests for each type individually.

## Figures and Tables

**Figure 1 molecules-27-04456-f001:**
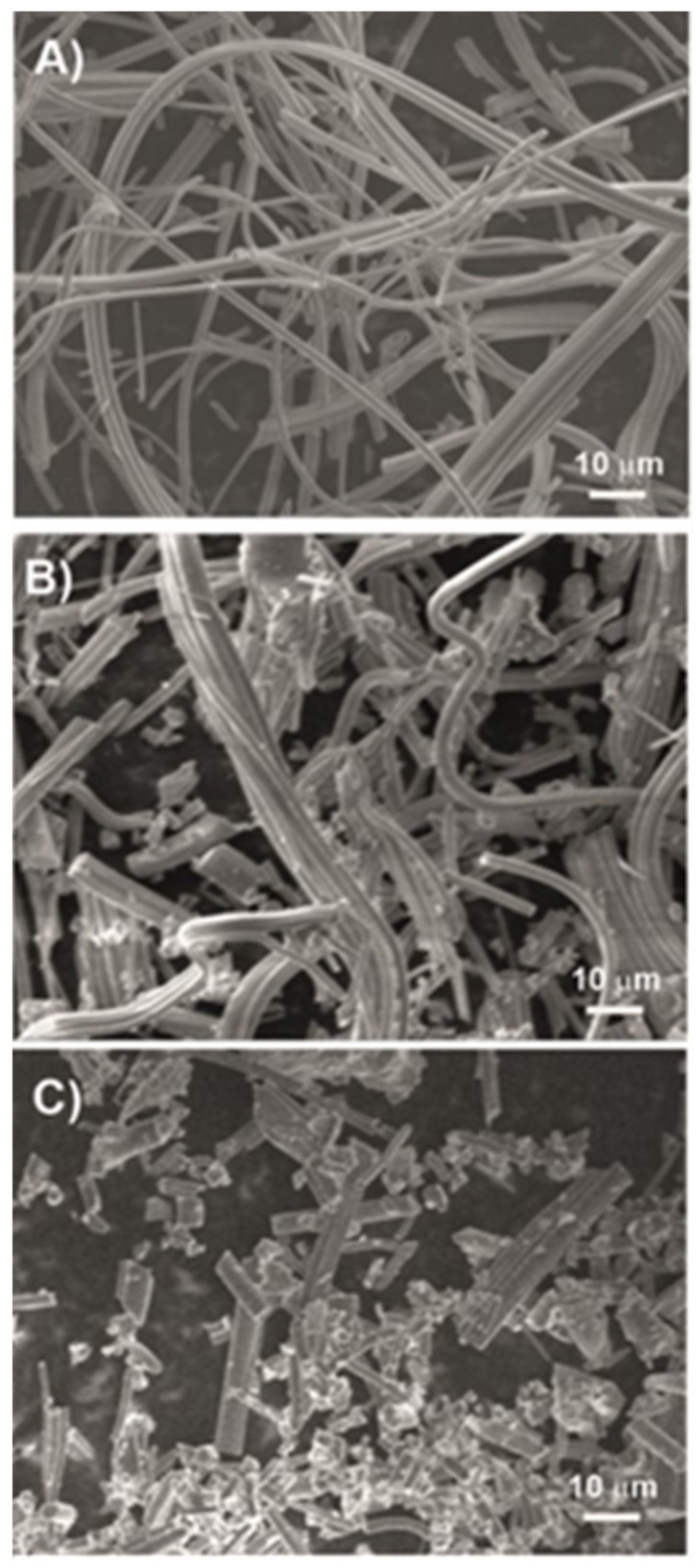
SEM images of (**A**) original inorganic fibers, (**B**) fibers milled for 5 s (L-SiO_2_NFs), and (**C**) fibers milled for 60 s (S-SiO_2_NFs).

**Figure 2 molecules-27-04456-f002:**
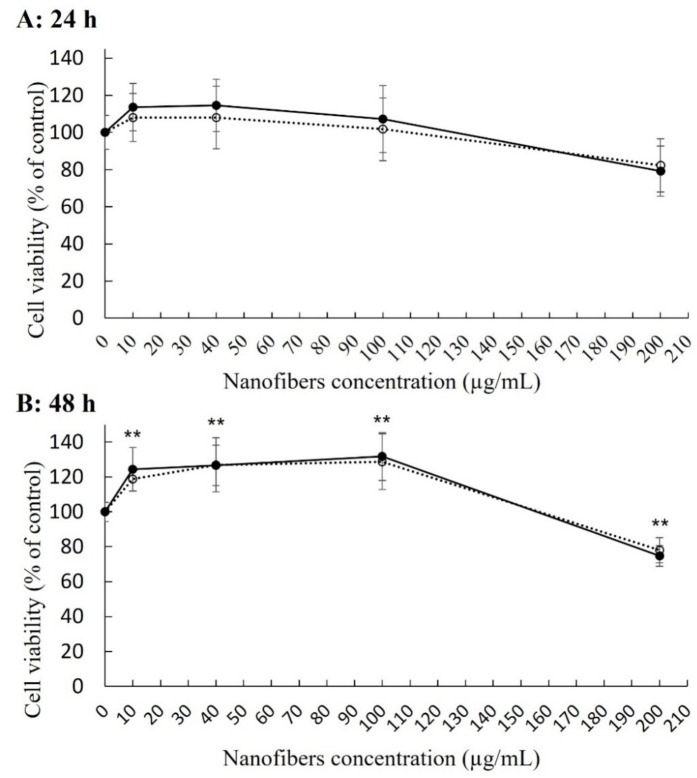
The effect of SiO_2_ nanofiber (SiO_2_NFs) lengths on the cytotoxic response. Cytotoxicity of L-SiO_2_NFs (○) or S-SiO_2_NFs (•) detected by WST-1 assay. Data were collected after 24 h (plot **A**) and 48 h (plot **B**) of treatment of THP-1-derived macrophage-like cells with different concentrations of SiO_2_NFs. Statistical analysis was performed by one-way analysis of variance, ** *p* = 0.005 (*n* > 6).

**Figure 3 molecules-27-04456-f003:**
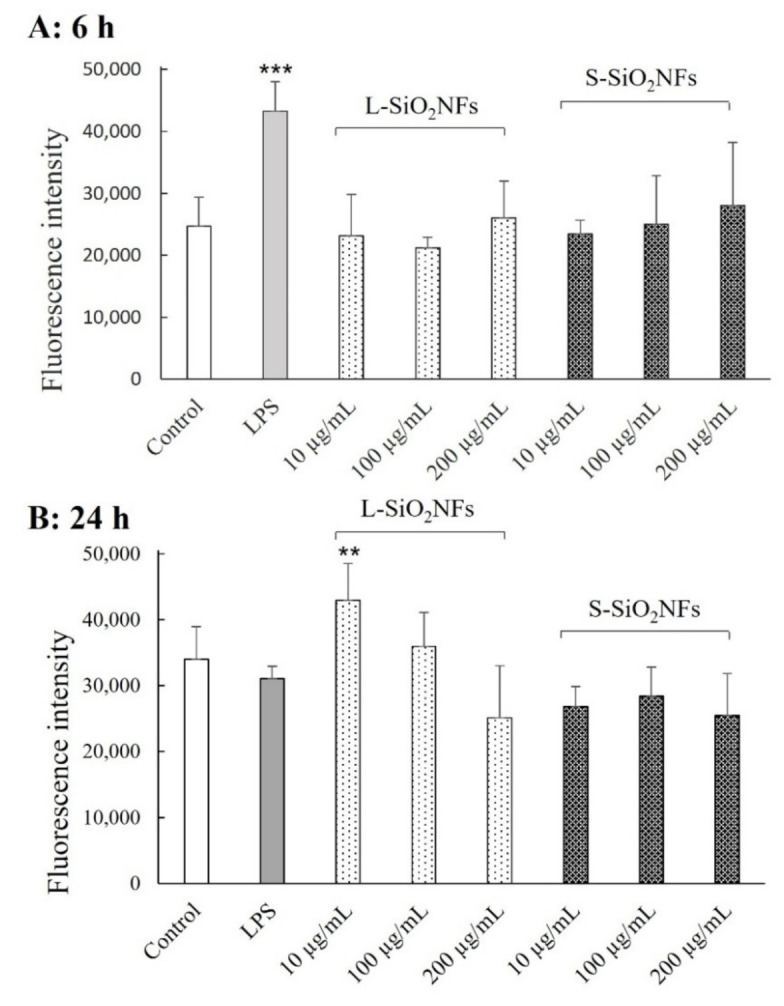
The effect of SiO_2_NF lengths on ROS production, detected by the H_2_DCFDA assay. THP-1 cells were seeded 3 × 10^4^/well and differentiated with PMA. THP-1-derived macrophage-like cells were treated with different SiO_2_NFs concentrations and 1 µg/mL LPS as the positive control for 6 (plot **A**) and 24 (plot **B**) hours. The assay was repeated 3 times, each time in triplicate. Statistical analysis was performed by one-way analysis of variance, *** *p* < 0.001 and ** *p* = 0.003 (*n* = 9).

**Figure 4 molecules-27-04456-f004:**
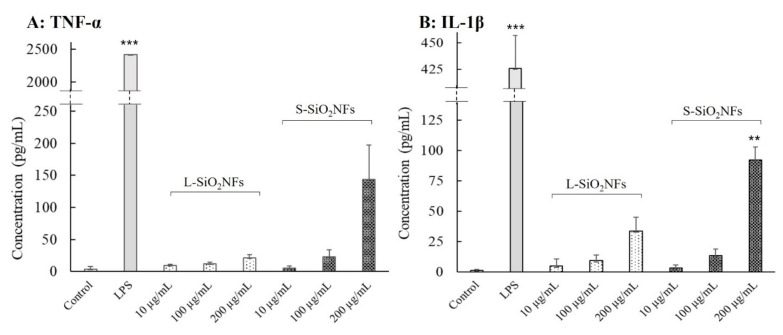
The effect of SiO_2_NF lengths on the cytokine expression. TNF-α (plot **A**) and IL-1β (plot **B**) productions were detected with the ELISA assay. THP-1 cells were seeded at 3 × 10^4^/well and differentiated with PMA. THP-1-derived macrophage-like cells were treated for 24 h with different SiO_2_NFs concentrations, and the supernatants were analyzed by ELISA according to the manufacturer’s instructions. The assay was repeated 4 times, each time in duplicate. Statistical analysis was performed by one-way analysis of variance, *** *p* < 0.001 and ** *p* = 0.006 (*n* = 8).

**Figure 5 molecules-27-04456-f005:**
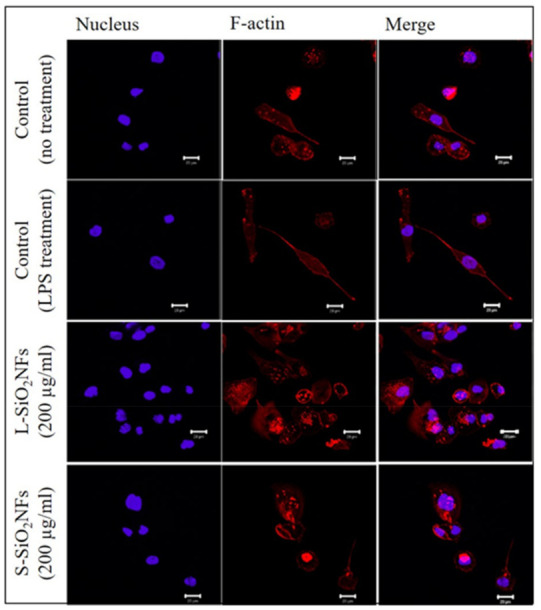
The effect of SiO_2_NF lengths on F-actin expression. THP-1-derived macrophage-like cells were stained with rhodamine-conjugated phalloidin (F-actin—red) and with DAPI (nucleus—blue) after 24 h of treatment with 200 µg/mL SiO_2_NFs. First row: nontreated control cells, second row: control cells treated with LPS, third row: after treatment with L-SiO_2_NFs, and fourth row: after treatment with S-SiO_2_NFs. Scale bar is 20 µm.

## Data Availability

The data presented in this study are available on request from the corresponding author. The raw/processed data required to reproduce these findings cannot be shared publicly at this time, as the data also form part of an ongoing study.

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
