# Peer review of "SiO2 Fibers of Two Lengths and Their Effect on Cellular Responses of Macrophage-like Cells"

_molecules, 2022, doi:10.3390/molecules27144456_

Round 1
Reviewer 1 Report
The length of SiO2 fibers affects cellular responses of macrophage-like cells
General Comments to Authors:
I would like to congratulate the authors for this understandable and clear manuscript.The manuscript is clearly written and well organized. The title is compatible with the basic content of the study.
The abstract section has sufficient information about the results of the study, however the aim/purpose of the research has not been clearly stated in the abstract.
The introduction section is short but contains the necessary information about the subject. Adding a few more new studies on the use of SiO2 nanofibers to this section will further strengthen the study. In this section, the sentence in the line 42 should be restated and written more clearly.
The results were given in a clear and wise order. But the authors should clarify that why did they chose the immunocompetent THP-1 cells? Figures are comprehensive and helpful. The lower doses (100 and 200 µg/mL) used in the study may be indicated in the Figure 1.
Why did the authors select 6h of treatment for ROS production induced by SiO2NFs? Writing a sentence of information about why LPS is chosen as a positive control can be informative for readers who do not know the subject. More citation and information is needed for compensatory cellular mechanisms. Because this is the mainstay of the authors, and it should be explained better. If it is thought that such an answer has been formed, the mechanism of this should be better explained and discussed.
In the discussion part the authors may be stated that the nontoxic effect of long SiO2NFs might be due to their inability to interact with the cells.The authors should refer to the previous publications about the impact of the SiO2 nanoparticle size and cellular toxicity. Internalization is important for cytotoxicity.
ROS production may be detected and visualised with fluorescent microscope. This type of result strengthen the quality of the work.
Reviewer 2 Report
The paper "The length of SiO2 fibers affects cellular responses of macrophage-like cells" by D. Smela et al. reported on the cytoxicity and inflammatory effects of SiO2 nanofibers on THP-1 cell line. The work presented here is well performed and the results supported the observations made.
I have some minor issues to be addressed:
1. The title refers to the lenght of SiO2 fiber even if there are only two length values reported here. I recommend to be changed.
2. In the Introduction the references assigned to the SiO2 particles are reduced. The authors should develop the field mainly highlighting the applicability of them in medicine, and thus establishing the advantages of the nanofiber use than the classical particles.
3.There is an allowable size for nanomaterials to be applied in biologic media. Is there an admissible dimension for large particles described here of 44 micrometers? These one prove to be involved in ROS production.
The authors also should improve the quality of the Figures 2 and 3.
To support the conclusions and also the title of the paper the authors must perform these studies on a wider class of dimensions of the fibers.
Taking into account the mentioned issues to be clarified I recommend the acceptance of this manuscript after Minor revision.
